# Prototypical Metric Transfer Learning for Continuous Speech Keyword Spotting With Limited Training Data

## Abstract

Continuous Speech Keyword Spotting (CSKS) is the problem of spotting keywords in recorded conversations, when a small number of instances of keywords are available in training data. Unlike the more common Keyword Spotting, where an algorithm needs to detect lone keywords or short phrases like *Alexa*, *Cortana*, *Hi Alexa!*, *Whatsup Octavia?* etc. in speech, CSKS needs to filter out embedded words from a continuous flow of speech, ie. spot *Anna* and *github* in *I know a developer named Anna who can look into this github issue*. Apart from the issue of limited training data availability, CSKS is an extremely imbalanced classification problem. We address the limitations of simple keyword spotting baselines for both aforementioned challenges by using a novel combination of loss functions (Prototypical networks loss and metric loss) and transfer learning. Our method improves F1 score by over 10%.

## 1 Introduction

Continuous Speech Keyword Spotting (CSKS) aims to detect embedded keywords in audio recordings. These spotted keyword frequencies can then be used to analyze theme of communication, creating temporal visualizations and word clouds clouds (2019). Another use case is to detect domain specific keywords which ASR (Automatic Speech Recognition) systems trained on public data cannot detect. For example, to detect a TV model number *W884* being mentioned in a recording, we might not have a large number of training sentences containing the model number of a newly launched TV to finetune a speech recognition (ASR) algorithm. A trained CSKS algorithm can be used to quickly extract out all instances of such keywords.

We train CSKS algorithms like other Keyword Spotting algorithms by classifying small fragments of audio in running speech. This requires the classifier model to have a formalized process to reject unseen instances (everything not a keyword, henceforth referred to as background) apart from ability to differentiate between classes (keywords). Another real world constraint that needs to be addressed while training such an algorithm is the availability of small amount of labeled keyword instances. We combine practices from fields of transfer learning, few-shot learning and metric learning to get better performance on this low training data imbalanced classification task.

Our work involves :

1. Testing existing Keyword Spotting methodologies Sainath & Parada (2015); Tang & Lin (2017) for the task of CSKS.

2. Proposing a transfer learning based baseline for CSKS by fine tuning weights of a publicly available deep ASR Hannun et al. (2014) model.

3. Introducing changes in training methodology by combining concepts from few-shot learning Snell et al. (2017) and metric learning Hoffer & Ailon (2015) into the transfer learning algorithm to address both the problems which baselines have a) missing keywords and b) false positives.

Our baselines, Honk(4.3.1), DeepSpeech-finetune(4.3.2), had comparatively both lower recall and precision. We noticed an improvement when fine tuning DeepSpeech model with prototypical loss

(DeepSpeech-finetune-prototypical (4.3.3)). While analysing the false positives of this model, it was observed that the model gets confused between the keywords and it also wrongly classifies background noise as a keyword. To improve this, we combined prototypical loss with a metric loss to reject background (DeepSpeech-finetune-prototypical+metric(4.3.4)). This model gave us the best results.

## 2 RELATED WORK

In the past, Hidden Markov Models (HMM) Weintraub (1993); Wilpon et al. (1990); Rose & Paul (1990) have been used to solve the CSKS problem. But since the HMM techniques use Viterbi algorithms (computationally expensive) a faster approach is required.

Owning to the popularity of deep learning, many recent works such as Lee et al. (2009); Mohamed et al. (2012); Grosse et al. (2012); Hinton et al. (2012); Dahl et al. (2012) have used deep learning techniques for many speech processing tasks. In tasks such as ASR, Hannun et al. Hannun et al. (2014) proposed a RNN based model to transcribe speech into text. Even for plain keyword spotting, Sainath & Parada (2015); Tang & Lin (2017); Li et al. (1992); Fernández et al. (2007); Chen et al. (2014); Pons et al. (2018) have proposed various deep learning architectures to solve the task. But to the best of our knowledge, no past work has deployed deep learning for spotting keywords in continuous speech.

Recently, a lot of work is being done on training deep learning models with limited training data. Out of them, few-shot techniques as proposed by Snell et al. (2017); Vinyals et al. (2016) have become really popular. Pons et al. Pons et al. (2018) proposed a few-shot technique using prototypical networks Snell et al. (2017) and transfer leaning Kunze et al. (2017); Choi et al. (2017) to solve a different audio task.

We took inspiration from these works to design our experiments to solve the CSKS task.

## 3 DATASET

Our learning data, which was created in-house, has 20 keywords to be spotted about television models of a consumer electronics brand. It was collected by making 40 participants utter each keyword 3 times. Each participant recorded in normal ambient noise conditions. As a result, after collection of learning data we have 120 (3 x 40) instances of each of the 20 keywords. We split the learning data 80:20 into train and validation sets. Train/Validation split was done on speaker level, so as to make sure that all occurrences of a particular speaker is present only on either of two sets. For testing, we used 10 different 5 minutes long simulated conversational recordings of television salesmen and customers from a shopping mall in India. These recordings contain background noise (as is expected in a mall) and have different languages (Indians speak a mixture of English and Hindi). The CSKS algorithm trained on instances of keywords in learning data is supposed to detect keywords embedded in conversations of test set.

## 4 APPROACH

### 4.1 DATA PREPROCESSING

Our dataset consisted of keyword instances but the algorithm trained using this data needs to classify keywords in fragments of running conversations. To address this, we simulate the continuous speech scenario, both for keyword containing audio and background fragments, by using publicly available audio data which consisted of podcasts audio, songs, and audio narration files. For simulating fragments with keywords, we extract two random contiguous chunks from these publicly available audio files and insert the keyword either in the beginning, in the middle or in the end of the chunks, thus creating an audio segment of 2 seconds. Random 2 second segments taken from publicly available audio are used to simulate segments with no keywords(also referred to as background elsewhere in the paper). These artificially simulated audio chunks from train/validation set of pure keyword utterances were used to train/validate the model. Since the test data is quite noisy, we further used various kinds of techniques such as time-shift, pitch-shift and intensity variation to

augment the data. Furthermore we used the same strategy as Tang et al. Tang & Lin (2017) of caching the data while training deep neural network on batches and artificially generating only 30% data which goes into a batch. By following these techniques, we could increase the data by many folds which not only helped the model to generalise better but also helped reduce the data preparation time during every epoch.

## 4.2 FEATURE ENGINEERING

For all the experiments using Honk architecture, MFCC features were used. To extract these features, 20Hz/4kHz band pass filters was used to reduce the random noise. Mel-Frequency Cepstrum Coefficient (MFCC) of forty dimension were constructed and stacked using 20 milliseconds window size with 10 miliseconds overlap. For all the experiments using deep speech architecture, we have extracted spectrograms of audio files using 20 milliseconds window size with 10 milliseconds overlap and 480 nfft value.

## 4.3 DEEP LEARNING ARCHITECTURES

### 4.3.1 HONK

Honk is a baseline Neural Network architecture we used to address the problem. Honk has shown good performance on normal Keyword Spotting and thus was our choice as the first baseline. The neural network is a Deep Residual Convolutional Neural Network He et al. (2016) which has number of feature maps fixed for all residual blocks. The python code of the model was taken from the open source repository of Convolutional Neural Networks for Keyword Spotting..

### 4.3.2 DEEPSPEECH-FINETUNE

DeepSpeech-finetune is fine tuning the weights of openly available DeepSpeech Hannun et al. (2014) model (initial feature extraction layers and not the final ASR layer) for CSKS task. The architecture consists of pretrained initial layers of DeepSpeech followed by a set of LSTM layers and a Fully Connected layer (initialized randomly) for classification. The output of Fully Connected layer is fed into a softmax and then a cross entropy loss for classification is used to train the algorithm. Please note that the finetune trains for 21 classes (20 keywords + 1 background) as in aforementioned Honk model.

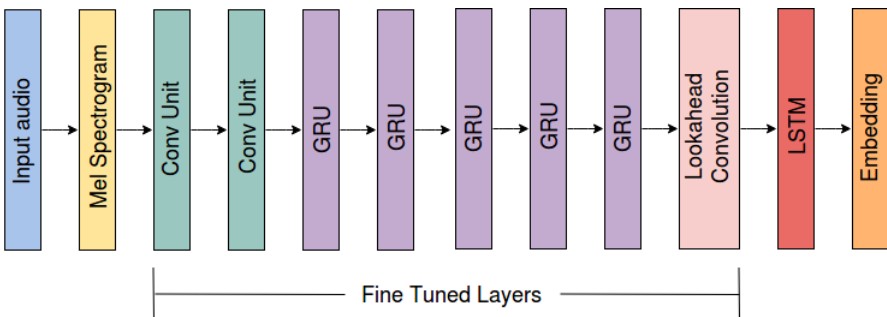

Figure 1: Architecture for DeepSpeech-finetune-prototypical

### 4.3.3 DEEPSPEECH-FINETUNE-PROTOTYPICAL

The next model we try is fine tuning DeepSpeech model but with a different loss function Snell et al. (2017). Prototypical loss works by concentrating embeddings of all data points of a class around the class prototype. This is done by putting a softmax on the negative distances from different prototypes to determine the probability to belong to corresponding classes. The architecture 1 is same as DeepSpeech-finetune, except output of pre-final layer is taken as embedding rather than applying a Fully Connected layer for classification. These embeddings are then used to calculate euclidean distances between datapoints and prototypes, represented as $d(embedding1, embedding2)$ in formulae. The softmax over negative distances from prototypes is used to train cross-entropy loss. During training, examples of each class are divided into support and query embeddings. The

Table 1: Results of all experiments

| Model | Recall | Precision | F1 |
|---|---|---|---|
| Honk | 0.46 | 0.34 | 0.39 |
| DeepSpeech-finetune | 0.267 | 0.244 | 0.256 |
| DeepSpeech-finetune-prototypical | 0.36 | 0.33 | 0.344 |
| DeepSpeech-finetune-prototypical+metric | **0.55** | **0.488** | **0.51** |

support embeddings are used to determine prototypes of the class. Equation 1 shows derivation of prototype of $k^{th}$ class where $f_\phi$ is the neural network yielding the embedding and $S_k$ is the set of support vectors for the class. The distance of query vectors from the prototypes of the class they belong to are minimized and prototypes of other classes is maximized when training the prototypical loss. The negative distances from the prototypes of each class are passed into softmax to get the probability of belonging in a class as shown in equation 2. We see better results when we train the algorithm using prototypical loss than normal cross entropy. On qualitatively observing the output from DeepSpeech-finetune-prototypical we see that the mistakes involving confusion between keywords are very less compared to datapoints of the class background being classified as one of the keywords. We hypothesize that this might be due to treating the entire background data as one class. The variance of background is very high and treating it as one class (a unimodal class in case of prototypes) might not be the best approach. To address this, we propose the next method where we use prototypes for classification within keywords and an additional metric loss component to keep distances of background datapoints from each prototype high.

$$\mathbf{c}_k = \frac{1}{S_k} \sum_{(\mathbf{x}_i, y_i) \in S_k} f_\phi(\mathbf{x}_i) \tag{1}$$

$$p(y = k | \mathbf{x}) = \frac{e^{-d(f_\phi(\mathbf{x}), \mathbf{c}_k)}}{\sum_j e^{-d(f_\phi(\mathbf{x}), \mathbf{c}_j)}} \tag{2}$$

### 4.3.4 DEEPSPEECH-FINETUNE-PROTOTYPICAL+METRIC

We hypothesize the components of loss function of this variant from failures of prototypical loss as stated earlier. The architecture is same as in 1, but the loss function is different from DeepSpeech-finetune-prototypical. While in DeepSpeech-finetune-prototypical, we trained prototype loss with 21 classes(20 keywords + 1 background), in DeepSpeech-finetune-prototypical+metric prototype loss is trained only amongst the 20 keywords and a new additional metric loss component inspired from Hoffer & Ailon (2015) is added to loss function. This metric loss component aims to bring datapoints of same class together and datapoints of different class further. Datapoints belonging to background are treated as different class objects for all other datapoints in a batch. So for each object in a batch, we add a loss component like equation 3 to prototypical loss. $\mathbf{c}^+$ is all datapoints in the batch belonging to the same class as $\mathbf{x}$ and $\mathbf{c}^-$ is all datapoints belonging to different classes than $\mathbf{x}$ (including background). This architecture gets the best results.

$$L_{metric} = \frac{e^{average(d(f_\phi(\mathbf{x}), \mathbf{c}^+))}}{e^{average(d(f_\phi(\mathbf{x}), \mathbf{c}^+))} + e^{average(d(f_\phi(\mathbf{x}), \mathbf{c}^-))}} \tag{3}$$

## 5 EXPERIMENTS, RESULTS AND DISCUSSION

While testing, the distance of a datapoint is checked with all the prototypes to determine its predicted class. Overlapping chunks of running audio are sent to the classifier to get classified for presence of a keyword. Train set numbers corresponding to all the models have shown in Table 1. DeepSpeech-finetune-prototypical+metric clearly beats the baselines in terms of both precision and recall. Honk is a respectable baseline and gets second best results after DeepSpeech-finetune-prototypical+metric, however, attempts to better Honk's performance using prototype loss and metric loss did not work at all. Our method to combine prototypical loss with metric learning can be used for any classification problem which has a set of classes and a large background class, but its effectiveness needs to be tested on other datasets.

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
