# OpenReview forum: "Prototypical Metric Transfer Learning for Continuous Speech Keyword Spotting With Limited Training Data"
_ICLR.cc/2019/Workshop/LLD — Submitted to LLD 2019_

### Official Review · AnonReviewer2 · 2019-03-31
**Reject**

**Rating:** 1
**Confidence:** 2

**Review:**

Summary:
The authors propose a method for detecting keywords in continuous speech given a limited number of training examples (120 for each keyword). A pre-trained ASR model is finetuned with a prototypical loss in conjunction with a metric loss. They find that their training method yields improved accuracy on an internal dataset.

Pros:
Considering the impact of continuous speech on keyword spotting is very important; I would go so far as to say that recognizing words spoken in isolation is not really keyword spotting. It's good to see that this is being considered in this paper.

Cons:
The claim that "since the HMM techniques use Viterbi algorithms (computationally expensive) a faster approach is required" is not well founded. This depends entirely on how many HMM states there are and how many possible states you can transition into. For example, Apple's "Hey Siri" detector runs the Viterbi algorithm on the outputs of a neural network for a super simple HMM where there are a small number of states and transitions are only allowed between the current phoneme and the next phoneme. They point out in their article about it that this part requires very little computation and almost all the computation is performed running the neural net (https://machinelearning.apple.com/2017/10/01/hey-siri.html).

The paper has a lot of stylistic bugs; consider the very second sentence: "These spotted keyword frequencies can then be used to analyze theme of communication, creating temporal visualizations and word clouds clouds (2019)." This type of thing happens all through the paper.

Weird, unnecessary details like "the python code of the model was taken from the open source repository of Convolutional Neural Networks for Keyword Spotting" are included. Unless you plan to release your code (?), this isn't really relevant.

The experiments are run only on an internal dataset, so it's impossible to replicate this paper.

---

### Official Review · AnonReviewer1 · 2019-04-08
**Good starting point, but needs more clarification**

**Rating:** 2
**Confidence:** 3

**Review:**

The task and method used for this paper seem quite promising, and the general task has broad applications as discussed in the paper and background sections. However, the paper seemed to trail off quickly after the first page, and the experiments section ended suddently. With some restructuring of the focus of the text to highlight the actual experiments and results, this paper could be much improved.

Details on the experiments were fairly limited - more discussion of the feature based frontend (were there standard delta and double delta with the mel, or only base mel?) and even some of the examples of the keywords in context would improve the work.

Rebalancing this work to include a more detailed analysis of the experiments, and reducing or minimizing the first page or so of content would be very helpful. As it stands, the experimental section is basically just the table, having more discussion of that result, and less of the surrounding description of the models, architecture and background would have been better in this 4 page format.

If the results are the focus of the paper, there should also be some testing done on standard benchmarks rather than only demonstrations on this custom dataset. If the focus is on the new dataset, there should be more discussion of the dataset structure, keywords, and some more exploratory analysis of the overall setting in order to motivate why the dataset is special, or different from standard benchmarks.

---

### Decision · Program_Chairs · 2019-04-08
**Acceptance Decision**

Reject